# Implicit Distributional Reinforcement Learning

**Yuguang Yue**[*], **Zhendong Wang**[*], **and Mingyuan Zhou**[†]
The University of Texas at Austin
Austin, TX 78712

## Abstract

To improve the sample efficiency of policy-gradient based reinforcement learning algorithms, we propose implicit distributional actor-critic (IDAC) that consists of a distributional critic, built on two deep generator networks (DGNs), and a semi-implicit actor (SIA), powered by a flexible policy distribution. We adopt a distributional perspective on the discounted cumulative return and model it with a state-action-dependent implicit distribution, which is approximated by the DGNs that take state-action pairs and random noises as their input. Moreover, we use the SIA to provide a semi-implicit policy distribution, which mixes the policy parameters with a reparameterizable distribution that is not constrained by an analytic density function. In this way, the policy's marginal distribution is implicit, providing the potential to model complex properties such as covariance structure and skewness, but its parameter and entropy can still be estimated. We incorporate these features with an off-policy algorithm framework to solve problems with continuous action space and compare IDAC with state-of-the-art algorithms on representative OpenAI Gym environments. We observe that IDAC outperforms these baselines in most tasks. Python code is provided[1].

## 1  Introduction

Model-free reinforcement learning (RL) plays an important role in addressing complex real-world sequential decision making tasks (MacAlpine and Stone, 2017; Silver et al., 2018; OpenAI, 2018). With the help of deep neural networks, model-free deep RL algorithms have been successfully implemented in a variety of tasks, including game playing (Silver et al., 2016; Mnih et al., 2013) and robotic control (Levine et al., 2016). Deep $Q$-network (DQN) (Mnih et al., 2015) enables RL agent with human level performance on Atari games (Bellemare et al., 2013), motivating many follow-up works with further improvements (Wang et al., 2016; Andrychowicz et al., 2017). A novel idea, proposed by Bellemare et al. (2017a), is to take a distributional perspective for deep RL problems, which models the full distribution of the discounted cumulative return of a chosen action at a state rather than just the expectation of it, so that the model can capture its intrinsic randomness instead of just first-order moment. Specifically, the distributional Bellman operator can help capture skewness and multimodality in state-action value distributions, which could lead to a more stable learning process, and approximating the full distribution may also mitigate the challenges of learning from a non-stationary policy. Under this distributional framework, Bellemare et al. (2017a) propose the C51 algorithm that outperforms previous state-of-the-art classical $Q$-learning based algorithms on a range of Atari games. However, some discrepancies exist between the theory and implementation in C51, motivating Dabney et al. (2018b) to introduce QR-DQN that borrows Wasserstein distance and quantile regression related techniques to diminish the theory-practice gap. Later on, the distributional view is also incorporated into the framework of deep deterministic policy gradient (DDPG) (Lillicrap et al., 2015) for continuous control tasks, yielding efficient algorithms such as distributed distributional

---

[*]The first two authors contributed equally. [†]Corresponding to `mingyuan.zhou@mccombs.utexas.edu`
[1]`https://github.com/zhougroup/IDAC`

DDPG (D4PG) (Barth-Maron et al., 2018) and sample-based distributional policy gradient (SDPG) (Singh et al., 2020). Due to the deterministic nature of the policy, these algorithms always manually add random noises to actions during the training process to avoid getting stuck in poor local optimums. By contrast, stochastic policy takes that randomness as part of the policy, learns it during the training, and achieves state-of-the-art performance, with the soft actor-critic (SAC) algorithm of Haarnoja et al. (2018) being a successful case in point.

Motivated by the promising directions from distributional action-value learning and stochastic policy, this paper integrates these two frameworks in hopes of letting them strengthen each other. We model the distribution of the discounted cumulative return of an action at a state with a deep generator network (DGN), whose input consists of a state-action pair and random noise, and applies the distributional Bellman equation to update its parameters. The DGN plays the role of a distributional critic, whose output conditioning on a state-action pair follows an implicit distribution. Intuitively, only modeling the expectation of the cumulative return is inevitably discarding useful information readily available during the training, and modeling the full distribution of it could capture more useful information to help better train and stabilize a stochastic policy. In other words, there are considerable potential gains in guiding the training of a distribution with a distribution rather than its expectation.

For stochastic policy, the default distribution choice under continuous control is diagonal Gaussian. However, assuming a unimodal and symmetric density at each dimension and independence between different dimensions make it incapable of capturing complex distributional properties, such as skewness, kurtosis, multimodality, and covariance structure. To fully take advantage of the distributional return modeled by the DGN, we thereby propose a semi-implicit actor (SIA) as the policy distribution, which adopts a semi-implicit hierarchical construction (Yin and Zhou, 2018) that can be made as complex as needed while remaining amenable to optimization via stochastic gradient descent (SGD). A naive combination of the DGN, an implicit distributional critic, and SIA, a semi-implicit actor, within an actor-critic policy gradient framework, however, only delivers mediocre performance, falling short of the promise it holds. We attribute its underachievement to the overestimation issue, commonly existing in classical value-based algorithms (Van Hasselt et al., 2016), that does not automatically go away under the distributional setting. Inspired by previous work in mitigating the overestimation issue in deep $Q$-learning (Fujimoto et al., 2018), we come up with a twin-delayed DGNs based critic, with which we provide a novel solution that takes the target values as the element-wise minimums of the sorted output values of these two DGNs, stabilizing the training process and boosting performance.

**Contributions:** The main contributions of this paper include: 1) we incorporate the distributional idea with the stochastic policy setting, and characterize the return distribution with the help of a DGN under a continuous control setup; 2) we introduce the twin-delayed structure on DGNs to mitigate the overestimation issue, involving element-wise minimization of two sorted vectors; and 3) we improve the flexibility of the policy by using a SIA instead of a Gaussian or mixture of Gaussian distribution to improve exploration, introducing an asymptotic lower bound for entropy estimation.

**Related work:** Since the successful implementation of RL problems from a distributional perspective on Atari 2600 games (Bellemare et al., 2017a), there is a number of follow-ups trying to boost existing deep RL algorithms by directly characterizing the distribution of the random return instead of the expectation (Dabney et al., 2018a,b; Barth-Maron et al., 2018; Singh et al., 2020). On the value-based side, C51 (Bellemare et al., 2017a) represents the return distribution with a categorical distribution defined by attaching $C = 51$ variable parameterized probabilities at $C = 51$ fixed locations. QR-DQN (Dabney et al., 2018b) does so by attaching $N$ variable parameterized locations at $N$ equally-spaced fixed quantiles, and employs a quantile regression loss for optimization. IQN (Dabney et al., 2018a) further extends this idea by learning a full quantile function. On the policy-gradient-based side, D4PG (Barth-Maron et al., 2018) incorporates the distributional perspective into DDPG (Lillicrap et al., 2015), with the return distribution modeled similarly as in C51. On top of that, SDPG of Singh et al. (2020) models the quantile function with a generator to overcome the limitation of using variable probabilities at fixed locations, and the same as D4PG, it models the policy as a deterministic transformation of the state representation. Though SDPG is applied under a deterministic setting, the quantile generator idea could be naturally extended to a stochastic policy setting.

There is rich literature aiming to obtain a high-expressive policy to encourage exploration during the training. When a deterministic policy is applied, a random perturb is always added when choosing a continuous action (Silver et al., 2014; Lillicrap et al., 2015). In Haarnoja et al. (2017), the policy is modeled proportional to its action-value function to guarantee flexibility. In Haarnoja

et al. (2018), SAC is proposed to mitigate the policy's expressiveness issue while retaining tractable optimization; with the policy modeled with either a Gaussian or a mixture of Gaussian, SAC adopts a maximum entropy RL objective function to encourage exploration. The normalizing flow (Rezende and Mohamed, 2015; Dinh et al., 2016) based techniques have been recently applied to design a flexible policy in both on-policy (Tang and Agrawal, 2018) and off-policy settings (Ward et al., 2019). To overcome the shortcomings of parametric policies, Tessler et al. (2019) propose "distributional" policy optimization that enhances the flexibility of the policy but still estimates the action-value function under a classical actor-critic setting. By contrast, IDAC estimates the action-value function under a "distributional" setting that directly models the distribution of the discounted cumulative return, a state-action dependent random variable whose expectation is the action-value function.

## 2 Implicit distributional actor-critic

We present implicit distributional actor-critic (IDAC) as a policy gradient based actor-critic algorithm under the off-policy learning setting, with a semi-implicit actor (SIA) and two deep generator networks (DGNs) as critics. We will start off with the introduction of distributional RL and DGN.

### 2.1 Implicit distributional RL with deep generator network (DGN)

We model the agent-environment interaction by a Markov decision process (MDP) denoted by $(\mathcal{S}, \mathcal{A}, R, P)$, where $\mathcal{S}$ is the state space, $\mathcal{A}$ the action space, $R$ a random reward function, and $P$ the environmental dynamics describing $P(s' \,|\, s, a)$, where $a \in \mathcal{A}$ and $s, s' \in \mathcal{S}$. A policy defines a map from the state space to action space $\pi(a \,|\, s) : \mathcal{S} \to \mathcal{A}$. Denote the discounted cumulative return from state-action pair $(s, a)$ following policy $\pi$ as $Z^\pi(s, a) = \sum_{t=0}^\infty \gamma^t R(s_t, a_t)$, where $\gamma$ is the discount factor, $s_0 := s$, and $a_0 := a$. Under a classic RL setting, an action-value function $Q$ is used to represent the expected return as $Q^\pi(s, a) = \mathbb{E}[Z^\pi(s, a)]$, where the expectation takes over all sources of intrinsic randomness (Goldstein et al., 1981). While under the distributional setup, it is the random return $Z^\pi(s, a)$ itself rather than its expectation that is being directly modeled. Similar to the classical Bellman equation, we have the *distributional Bellman equation* (Dabney et al., 2018b) as

$$Z^\pi(s, a) \overset{D}{=} R(s, a) + \gamma Z^\pi(s', a'). \tag{1}$$

where $\overset{D}{=}$ denotes "equal in distribution" and $a' \sim \pi(\cdot \,|\, s')$, $s' \sim P(\cdot \,|\, s, a)$.

We propose using a DGN to model the distribution of random return $Z^\pi$ as

$$Z^\pi(s, a) \overset{D}{\approx} G_{\boldsymbol{\omega}}(s, a, \boldsymbol{\epsilon}), \ \boldsymbol{\epsilon} \sim p(\boldsymbol{\epsilon}), \tag{2}$$

where $\overset{D}{\approx}$ denotes "approximately equal in distribution," $p(\boldsymbol{\epsilon})$ is a random noise distribution, and $G_{\boldsymbol{\omega}}(s, a, \boldsymbol{\epsilon})$ is a neural network based deterministic function parameterized by $\boldsymbol{\omega}$, whose input consists of $s$, $a$, and $\boldsymbol{\epsilon}$. We can consider $G_{\boldsymbol{\omega}}(s, a, \boldsymbol{\epsilon})$ as a generator that transforms $p(\boldsymbol{\epsilon})$ into an implicit distribution, from which random samples can be straightforwardly generated but the probability density function is in general not analytic (*e.g.*, when $G_{\boldsymbol{\omega}}(s, a, \boldsymbol{\epsilon})$ is not invertible with respect to $\boldsymbol{\epsilon}$). If the distributional equality holds in (2), we can approximate the distribution of $Z^\pi(s, a)$ in a sample-based manner, which can be empirically represented by $K$ independent, and identically distributed (*iid*) random samples as $\{G_{\boldsymbol{\omega}}(s, a, \boldsymbol{\epsilon}_1), \cdots, G_{\boldsymbol{\omega}}(s, a, \boldsymbol{\epsilon}_K)\}$, where $\boldsymbol{\epsilon}_1, \ldots, \boldsymbol{\epsilon}_K \overset{iid}{\sim} p(\boldsymbol{\epsilon})$.

### 2.2 Learning of DGN

Based on (1), we desire the DGN to also satisfy the distributional matching that

$$G_{\boldsymbol{\omega}}(s, a, \boldsymbol{\epsilon}) \overset{D}{=} R(s, a) + \gamma G_{\boldsymbol{\omega}}(s', a', \boldsymbol{\epsilon}'), \quad \text{where } \boldsymbol{\epsilon}, \boldsymbol{\epsilon}' \overset{iid}{\sim} p(\boldsymbol{\epsilon}). \tag{3}$$

This requires us to adopt a differential metric to measure the distance between two distributions and use it to guide the learning of the generator parameter $\boldsymbol{\omega}$. While there exist powerful methods to learn high-dimensional data generators, such as generative adverserial nets (Goodfellow et al., 2014; Arjovsky et al., 2017), there is no such need here as there exist simple and stable solutions to estimate the distance between two one-dimensional distributions given *iid* random samples from them.

In particular, the $p$-Wasserstein distance (Villani, 2008) between the distributions of univariate random variables $X, Y \in \mathbb{R}$ can be approximated by that between their empirical distributions supported on

$K$ random samples, expressed as $\hat{X} = \frac{1}{K}\sum_{k=1}^{K}\delta_{x_k}$ and $\hat{Y} = \frac{1}{K}\sum_{k=1}^{K}\delta_{y_k}$, and we have

$$W_p(X,Y)^p \approx W_p(\hat{X},\hat{Y})^p = \frac{1}{K}\sum_{k=1}^{K}||\overrightarrow{x}_k - \overrightarrow{y}_k||^p, \tag{4}$$

where $\overrightarrow{x}_{1:K}$ is obtained by sorting the $K$ elements in $x_{1:K}$ in increasing order and $\overrightarrow{y}_{1:K}$ is obtained from $y_{1:K}$ in the same manner (Villani, 2008; Bernton et al., 2019; Deshpande et al., 2018; Kolouri et al., 2019). Though seems tempting to use $W_p(\hat{X},\hat{Y})^p$ as the loss function, it has been shown (Bellemare et al., 2017b; Dabney et al., 2018b) that such a loss function may not be theoretically sound when optimized with SGD, motivating the use of a quantile regression loss based on $\hat{X}$ and $\hat{Y}$. In addition, it is unclear whether the empirical samples based estimation of the Wasserstein distance shown in (4) remains sound in theory if the $L_p$ norm is replaced by the Huber loss (Huber, 1992). To this end, we propose to generalize the method in Dabney et al. (2018b) to measure the distributional distance with a quantile regression Huber loss based on empirical samples; different from Dabney et al. (2018b) who use a deep NN to estimate the action values at fixed quantile locations for each $(\boldsymbol{s},\boldsymbol{a})$, providing no guarantee that the NN output at a designated higher quantile is larger than that at a lower quantile, there is no such concern in DGN that simply sorts its $iid$ sampled values to define a quantile regression Huber loss as

$$\mathcal{L}_{\text{QR}}(X,Y) \approx \mathcal{L}_{\text{QR}}(\hat{X},\hat{Y}) = \frac{1}{K^2}\sum_{k=1}^{K}\sum_{k'=1}^{K}\rho_{\tau_k}^{\kappa}(y_{k'} - \overrightarrow{x}_k), \tag{5}$$

where $\overrightarrow{x}_k$ that are arranged in increasing order are one-to-one mapped to $K$ equally-spaced increasing quantiles $\tau_k = (k-0.5)/K$, $\kappa$ is a pre-fixed threshold (set as $\kappa = 1$ unless specified otherwise), and

$$\rho_{\tau_k}^{\kappa}(u) = |\tau_k - \mathbf{1}_{[u<0]}|\mathcal{L}_{\kappa}(u)/\kappa, \quad \mathcal{L}_{\kappa}(u) = \frac{1}{2}u^2\mathbf{1}_{[|u|\leq\kappa]} + \kappa(|u| - \frac{1}{2}\kappa)\mathbf{1}_{[|u|>\kappa]}. \tag{6}$$

Note that the reason we map $\overrightarrow{x}_k$ to quantile $\tau_k = (k-0.5)/K$, for $k = 1,\ldots,K$, is because $P(X \leq \overrightarrow{x}_k) \approx \tau_k$, an approximation that becomes increasingly more accurate as $K$ increases.

Recall the distributional matching objective in (3). To train the DGN, we first obtain an empirical distribution $\hat{X}$ of the generator supported on $K$ $iid$ random samples as

$$x_{1:K} := \{G_{\boldsymbol{\omega}}(\boldsymbol{s},\boldsymbol{a},\boldsymbol{\epsilon}^{(k)})\}_{1:K}, \quad \text{where } \boldsymbol{\epsilon}^{(1)},\ldots,\boldsymbol{\epsilon}^{(K)} \overset{iid}{\sim} p(\boldsymbol{\epsilon}), \tag{7}$$

and similarly an empirical target distribution $\hat{Y}$ supported on

$$y_{1:K} := \{R(\boldsymbol{s},\boldsymbol{a}) + \gamma G_{\tilde{\boldsymbol{\omega}}}(\boldsymbol{s}',\boldsymbol{a}',\boldsymbol{\epsilon}'^{(k)})\}_{1:K}, \quad \text{where } \boldsymbol{\epsilon}'^{(1)},\ldots,\boldsymbol{\epsilon}'^{(K)} \overset{iid}{\sim} p(\boldsymbol{\epsilon}), \tag{8}$$

where $\boldsymbol{a}' \sim \pi(\cdot\,|\,\boldsymbol{s}')$, $\boldsymbol{s}' \sim P(\cdot\,|\,\boldsymbol{s},\boldsymbol{a})$, and $\tilde{\boldsymbol{\omega}}$ is the delayed generator parameter, a common practice to stabilize the learning process as used in Lillicrap et al. (2015) and Fujimoto et al. (2018). Since we use empirical samples to represent the distributions, we first sort $x_{1:K}$ in increasing order, denoted as

$$(\overrightarrow{x}_1,\cdots,\overrightarrow{x}_K) = \text{sort}(x_1,\cdots,x_K),$$

and then map them to increasing quantiles $((k-0.5)/K)_{1:K}$. The next step is to minimize the quantile regression Huber loss as in (5), and the objective function for DGN parameter $\boldsymbol{\omega}$ becomes

$$J(\boldsymbol{\omega}) = \mathcal{L}_{\text{QR}}(\hat{X}, \text{StopGradient}\{\hat{Y}\}) = \frac{1}{K^2}\sum_{k=1}^{K}\sum_{k'=1}^{K}\rho_{\tau_k}^{\kappa}(\text{StopGradient}\{y_{k'}\} - \overrightarrow{x}_k). \tag{9}$$

## 2.3 Twin delayed DGNs

Motivated by the significant improvement shown in Fujimoto et al. (2018), we propose the use of twin DGNs to prevent overestimation of the return distribution. However, it cannot be applied directly. On value-based algorithm, one can directly take the minimum of two estimated $Q$-values; on the other hand, we have empirical samples from a distribution and we try to avoid overestimation on that distribution which needs to be taken care of. Specifically, we design two DGNs $G_{\boldsymbol{\omega}_1}(\boldsymbol{s},\boldsymbol{a},\boldsymbol{\epsilon})$ and $G_{\boldsymbol{\omega}_2}(\boldsymbol{s},\boldsymbol{a},\boldsymbol{\epsilon})$ with independent initialization of $\boldsymbol{\omega}_1$ and $\boldsymbol{\omega}_2$ and independent input noise. Therefore, we will have two sets of target values as defined in (8), which are denoted as $y_{1,1:K}$ and $y_{2,1:K}$, respectively. Since they represent empirical distributions now and each element of them is assigned to one specific quantile, we will need to sort them before taking element-wise minimum so that the distribution is not distorted before mitigating the overestimation issue. In detail, with

$$(\overrightarrow{y}_{1,1},\cdots,\overrightarrow{y}_{1,K}) = \text{sort}(y_{1,1},\cdots,y_{1,K}), \quad (\overrightarrow{y}_{2,1},\cdots,\overrightarrow{y}_{2,K}) = \text{sort}(y_{2,1},\cdots,y_{2,K}),$$

the new target values for twin DGNs become

$$(\overrightarrow{y}_1,\cdots,\overrightarrow{y}_K) = \left(\min(\overrightarrow{y}_{1,1},\overrightarrow{y}_{2,1}),\cdots,\min(\overrightarrow{y}_{1,K},\overrightarrow{y}_{2,K})\right),$$

and with $\boldsymbol{\epsilon}^{(1)},\ldots,\boldsymbol{\epsilon}^{(K)} \overset{iid}{\sim} p(\boldsymbol{\epsilon})$, the objective function for parameter $\boldsymbol{\omega}_1$ of twin DGNs becomes

$$J(\boldsymbol{\omega}_1) = \frac{1}{K^2}\sum_{k=1}^{K}\sum_{k'=1}^{K}\rho_{\tau_k}^{\kappa}(\text{StopGradient}\{\overrightarrow{y}_{k'}\} - \overrightarrow{x}_k), \quad x_{1:K} := \{G_{\boldsymbol{\omega}_1}(\boldsymbol{s},\boldsymbol{a},\boldsymbol{\epsilon}^{(k)})\}_{1:K}. \tag{10}$$

The objective function for parameter $\boldsymbol{\omega}_2$ is similarly defined under the same set of target values.

## 2.4 Semi-implicit actor (SIA)

Since the return distribution is modeled in a continuous action space, it will be challenging to choose the action that maximizes the critic. We instead turn to finding a flexible stochastic policy that captures the energy landscape of $\mathbb{E}_{\boldsymbol{\epsilon} \sim p(\boldsymbol{\epsilon})}[G_{\boldsymbol{\omega}}(\boldsymbol{s}, \boldsymbol{a}, \boldsymbol{\epsilon})]$. The default parametric policy for continuous control problems is modeled as a diagonal Gaussian distribution, where the means and variances of all dimensions are obtained from some deterministic transformations of state $\boldsymbol{s}$. Due to the nature of the diagonal Gaussian distribution, it can not capture the dependencies between different action dimensions and has a unimodal and symmetric assumption on its density function at each dimension, limiting its ability to encourage exploration. For example, it may easily get stuck in a bad local mode simply because of its inability to accomodate multi-modality (Yue et al., 2020).

To this end, we consider a semi-implicit construction (Yin and Zhou, 2018) that enriches the diagonal Gaussian distribution by randomizing its parameters with another distribution, making the marginal of the semi-implicit hierarchy, which in general has no analytic density function, become capable of modeling much more complex distributional properties, such as skewness, multi-modality, and dependencies between different action dimensions. In addition, its parameters are amenable to SGD based optimization, making it even more attractive as a plug-in replacement of diagonal Gaussian. Specifically, we construct a semi-implicit policy with a hierarchical structure as

$$\pi_{\boldsymbol{\theta}}(\boldsymbol{a} \,|\, \boldsymbol{s}) = \int_{\boldsymbol{\xi}} \pi_{\boldsymbol{\theta}}(\boldsymbol{a} \,|\, \boldsymbol{s}, \boldsymbol{\xi}) p(\boldsymbol{\xi}) d\boldsymbol{\xi}, \;\; \text{where } \pi_{\boldsymbol{\theta}}(\boldsymbol{a} \,|\, \boldsymbol{s}, \boldsymbol{\xi}) = \mathcal{N}(\boldsymbol{a}; \boldsymbol{\mu}_{\boldsymbol{\theta}}(\boldsymbol{s}, \boldsymbol{\xi}), \text{diag}\{\boldsymbol{\sigma}_{\boldsymbol{\theta}}^2(\boldsymbol{s}, \boldsymbol{\xi})\}), \quad (11)$$

where $\boldsymbol{\theta}$ denotes the policy parameter and $\boldsymbol{\xi} \sim p(\boldsymbol{\xi})$ denotes a random noise, which concatenated with state $\boldsymbol{s}$ is transformed by a deep neural network parameterized by $\boldsymbol{\theta}$ to define both the mean and covariance of a diagonal Gaussian policy distribution. Note while we choose $\pi_{\boldsymbol{\theta}}(\boldsymbol{a} \,|\, \boldsymbol{s}, \boldsymbol{\xi})$ to be diagonal Gaussian, it can take any explicit reparameterizable distribution. There is no constraint on $p(\boldsymbol{\xi})$ as long as it is simple to sample from, and is reparameterizable if it contains parameters to learn. This semi-implicit construction balances the tractability and expressiveness of $\pi_{\boldsymbol{\theta}}(\boldsymbol{a} \,|\, \boldsymbol{s})$, where we can get a powerful implicit policy while still be capable of sampling from it and estimating its entropy. Based on previous proofs (Yin and Zhou, 2018; Molchanov et al., 2019), we present the following Lemma for entropy estimation and defer its proof to the Appendix. The ability of entropy estimation is crucial when solving problems under the maximum entropy RL framework (Todorov, 2007; Ziebart, 2010; Ziebart et al., 2008), which we adopt below to encourage exploration.

**Lemma 1.** *Assume $\pi_{\boldsymbol{\theta}}(\boldsymbol{a} \,|\, \boldsymbol{s})$ is constructed as in Eq. (11), the following expectation*

$$\mathcal{H}_L := \mathbb{E}_{\boldsymbol{\xi}^{(0)}, \ldots, \boldsymbol{\xi}^{(L)} \overset{iid}{\sim} p(\boldsymbol{\xi})} \mathbb{E}_{\boldsymbol{a} \sim \pi_{\boldsymbol{\theta}}(\boldsymbol{a} \,|\, \boldsymbol{s}, \boldsymbol{\xi}^{(0)})}[\log \tfrac{1}{L+1} \sum_{\ell=0}^{L} \pi_{\boldsymbol{\theta}}(\boldsymbol{a} \,|\, \boldsymbol{s}, \boldsymbol{\xi}^{(\ell)})] \tag{12}$$

*is an asymptotically tight upper bound of the negative entropy, expressed as*

$$\mathcal{H}_{\ell} \geq \mathcal{H}_{\ell+1} \geq \mathcal{H} := \mathbb{E}_{\boldsymbol{a} \sim \pi_{\boldsymbol{\theta}}(\boldsymbol{a} \,|\, \boldsymbol{s})}[\log \pi_{\boldsymbol{\theta}}(\boldsymbol{a} \,|\, \boldsymbol{s})], \quad \forall \ell \geq 0.$$

## 2.5 Learning of SIA

In IDAC, the action-value function can be expressed as $\mathbb{E}_{\boldsymbol{\epsilon}}[G_{\boldsymbol{\omega}}(\boldsymbol{s}, \boldsymbol{a}, \boldsymbol{\epsilon})]$. Related to SAC (Haarnoja et al., 2018), we learn the policy towards the Boltzman distribution of the action-value function by minimizing a Kullback–Leibler (KL) divergence between them as

$$\pi_{\text{new}} = \text{argmin}_{\pi' \in \boldsymbol{\Pi}} \mathbb{E}_{\boldsymbol{s} \sim \rho(\boldsymbol{s})} \left[ \text{KL}\left( \pi_{\boldsymbol{\theta}}(\boldsymbol{a} \,|\, \boldsymbol{s}) \middle\| \frac{\exp(\mathbb{E}_{\boldsymbol{\epsilon} \sim p(\boldsymbol{\epsilon})}[G(\boldsymbol{s}, \boldsymbol{a}, \boldsymbol{\epsilon})/\alpha])}{\int \exp(\mathbb{E}_{\boldsymbol{\epsilon} \sim p(\boldsymbol{\epsilon})}[G(\boldsymbol{s}, \boldsymbol{a}, \boldsymbol{\epsilon})/\alpha]) d\boldsymbol{a}} \right) \right], \tag{13}$$

where $\rho(\boldsymbol{s})$ denotes the state-visitation frequency, $\alpha > 0$ is a reweard scaling coefficient, and $\boldsymbol{\Pi}$ is the semi-implicit distribution family. Therefore, the loss function for policy parameters is

$$J(\boldsymbol{\theta}) = -\mathbb{E}_{\boldsymbol{s} \sim \rho(\boldsymbol{s})} \mathbb{E}_{\boldsymbol{a} \sim \pi_{\boldsymbol{\theta}}(\cdot \,|\, \boldsymbol{s})} \{\mathbb{E}_{\boldsymbol{\epsilon} \sim p(\boldsymbol{\epsilon})}[G_{\boldsymbol{\omega}}(\boldsymbol{s}, \boldsymbol{a}, \boldsymbol{\epsilon})] - \alpha \log \pi_{\boldsymbol{\theta}}(\boldsymbol{a} \,|\, \boldsymbol{s})\}. \tag{14}$$

We cannot optimize (14) directly since as the semi-implicit policy $\pi_{\boldsymbol{\theta}}(\boldsymbol{a} \,|\, \boldsymbol{s})$ does not have an analytic density function and its entropy is not analytic. With the help of Lemma 1, we turn to minimizing an asymptotic upper bound of (14) as

$$J(\boldsymbol{\theta}) \leq \mathbb{E}_{\boldsymbol{s} \sim \rho(\boldsymbol{s})} \mathbb{E}_{\boldsymbol{\xi}^{(1)}, \ldots, \boldsymbol{\xi}^{(L)} \overset{iid}{\sim} p(\boldsymbol{\xi})} \tfrac{1}{J} \sum_{j=1}^{J} \mathbb{E}_{\boldsymbol{\xi}_j^{(0)} \sim p(\boldsymbol{\xi})} \mathbb{E}_{\boldsymbol{a}^{(j)} \sim \pi_{\boldsymbol{\theta}}(\cdot \,|\, \boldsymbol{s}, \boldsymbol{\xi}_j^{(0)})} \mathbb{E}_{\boldsymbol{\epsilon}^{(j)} \sim p(\boldsymbol{\epsilon})}[\overline{J}_j(\boldsymbol{\theta})]$$

$$\overline{J}_j(\boldsymbol{\theta}) := -\left( \tfrac{1}{2} \sum_{i=1}^{2} G_{\boldsymbol{\omega}_i}(\boldsymbol{s}, \boldsymbol{a}^{(j)}, \boldsymbol{\epsilon}^{(j)}) \right) + \alpha \log \left( \frac{\pi_{\boldsymbol{\theta}}(\boldsymbol{a}^{(j)} \,|\, \boldsymbol{s}, \boldsymbol{\xi}_j^{(0)}) + \sum_{\ell=1}^{L} \pi_{\boldsymbol{\theta}}(\boldsymbol{a}^{(j)} \,|\, \boldsymbol{s}, \boldsymbol{\xi}^{(\ell)})}{L+1} \right), \tag{15}$$

**Algorithm 1** IDAC: Implicit Distributional Actor-Critic (see Appendix B for more implementation details)
***

**Require:** Learning rate $\lambda$, smoothing factor $\tau$. Initial policy network parameter $\boldsymbol{\theta}$, distributional generator network parameters $\boldsymbol{\omega}_1, \boldsymbol{\omega}_2$, entropy coefficient $\eta$;

$\tilde{\boldsymbol{\omega}}_1 \leftarrow \boldsymbol{\omega}_1, \tilde{\boldsymbol{\omega}}_2 \leftarrow \boldsymbol{\omega}_2, \mathcal{D} \leftarrow \emptyset$

**for** Each iteration **do**

    **for** Each environment step **do**

        $\boldsymbol{\xi}_t \sim p(\boldsymbol{\xi})$, $\boldsymbol{a}_t \sim \pi_{\boldsymbol{\theta}}(\cdot \mid \boldsymbol{s}_t, \boldsymbol{\xi}_t)$ {Sample noise and then action}

        $\boldsymbol{s}_{t+1} \sim p(\cdot \mid \boldsymbol{s}_t, \boldsymbol{a}_t)$ {Observe next state}

        $\mathcal{D} \leftarrow \mathcal{D} \cup (\boldsymbol{s}_t, \boldsymbol{a}_t, r_t, \boldsymbol{s}_{t+1})$ {Store transition tuples}

    **end for**

    Sample transitions from the replay buffer

    $\boldsymbol{\omega}_i \leftarrow \boldsymbol{\omega}_i - \lambda \nabla_{\boldsymbol{\omega}_i} J(\boldsymbol{\omega}_i)$ for $i = 1, 2$ {Update DGNs, Eq. (10)}

    $\boldsymbol{\theta} \leftarrow \boldsymbol{\theta} - \lambda \nabla_{\boldsymbol{\theta}} \overline{J}(\boldsymbol{\theta})$ {Update SIA, Eq. (16)}

    $\eta \leftarrow \eta - \lambda \nabla_{\eta} J(\eta)$, let $\alpha = \exp(\eta)$ {Update entropy coefficient, Eq. (17)}

    $\tilde{\boldsymbol{\omega}}_i \leftarrow \tau \boldsymbol{\omega}_i + (1 - \tau) \tilde{\boldsymbol{\omega}}_i$ for $i = 1, 2$ {Soft update delayed networks}

**end for**
***

where $\boldsymbol{\omega}_1, \boldsymbol{\omega}_2$ are the parameters of twin DGNs, $\overline{J}_j(\theta)$ is a Monte Carlo estimate of this asymptotic upper bound given a single action, and $J$ is the number of actions that we will use to estimate the objective function. Note we could set $J = 1$, but then we will still need to sample multiple *iid* $\boldsymbol{\epsilon}$'s to estimate the action-value function. An alternative choice is to sample $J > 1$ actions and sample multiple $\boldsymbol{\epsilon}$'s for each action, which, given the same amount of computational budget, is in general found to be less efficient than simply increasing the number of actions in (15). To estimate the gradient, each $\boldsymbol{a}^{(j)} \sim \pi_{\boldsymbol{\theta}}(\boldsymbol{a} \mid \boldsymbol{s}, \boldsymbol{\xi}_j^{(0)})$ is sampled via the reparametrization trick by letting $\boldsymbol{a}^{(j)} = \mathcal{T}_{\boldsymbol{\theta}}(\boldsymbol{s}, \boldsymbol{\xi}_j^{(0)}, \boldsymbol{e}_j)$, $\boldsymbol{e}_j \sim p(\boldsymbol{e})$ to ensure low gradient estimation variance, which means it is deterministically transformed from $\boldsymbol{s}, \boldsymbol{\xi}_j^{(0)}$, and random noise $\boldsymbol{e}_j \sim p(\boldsymbol{e})$ with a nueral network parameterized by $\boldsymbol{\theta}$. To compute the gradient of $\overline{J}(\boldsymbol{\theta}) := \sum_{j=1}^{J} \overline{J}_j(\boldsymbol{\theta})$ with respect to $\boldsymbol{\theta}$, we notice that $\nabla_{\boldsymbol{\theta}} \log \left( \frac{\pi_{\boldsymbol{\theta}}(\boldsymbol{a}^{(j)} \mid \boldsymbol{s}, \boldsymbol{\xi}_j^{(0)}) + \sum_{\ell=1}^{L} \pi_{\boldsymbol{\theta}}(\boldsymbol{a}^{(j)} \mid \boldsymbol{s}, \boldsymbol{\xi}^{(\ell)})}{L+1} \right)$ can be rewritten as the summation of two terms: the first term is obtained by treating $\boldsymbol{a}^{(j)}$ in $\pi_{\boldsymbol{\theta}}(\boldsymbol{a}^{(j)} \mid -)$ as constants, and the second term by treating $\boldsymbol{\theta}$ in $\pi_{\boldsymbol{\theta}}(\cdot)$ as constants. Since $\mathbb{E}_{\boldsymbol{a} \sim \pi_{\boldsymbol{\theta}}(\boldsymbol{a} \mid \boldsymbol{s})}[\nabla_{\boldsymbol{\theta}} \log \pi_{\boldsymbol{\theta}}(\boldsymbol{a} \mid \boldsymbol{s})] = 0$, the expectation of the first term becomes zero when $L \to \infty$. For this reason, we omit its contribution to the gradient when computing $\nabla_{\boldsymbol{\theta}} \overline{J}(\boldsymbol{\theta})$, which can then be expressed as

$$\nabla_{\boldsymbol{\theta}} \overline{J}(\boldsymbol{\theta}) = -\sum_{j=1}^{J} \left\{ \left[ \left( \frac{1}{2J} \sum_{i=1}^{2} \nabla_{\boldsymbol{a}^{(j)}} G_{\boldsymbol{\omega}_i}(\boldsymbol{s}, \boldsymbol{a}^{(j)}, \boldsymbol{\epsilon}^{(j)}) \right) \right. \right.$$

$$\left. \left. -\frac{1}{J}\alpha \sum_{\ell=0}^{L} \frac{\pi_{\boldsymbol{\theta}}(\boldsymbol{a}^{(j)} \mid \boldsymbol{s}, \boldsymbol{\xi}_j^{(\ell)})}{\sum_{\ell'=0}^{L} \pi_{\boldsymbol{\theta}}(\boldsymbol{a}^{(j)} \mid \boldsymbol{s}, \boldsymbol{\xi}_j^{(\ell')})} \nabla_{\boldsymbol{a}^{(j)}} \log \pi_{\boldsymbol{\theta}}(\boldsymbol{a}^{(j)} \mid \boldsymbol{s}, \boldsymbol{\xi}_j^{(\ell)}) \right] \Big|_{\boldsymbol{a}^{(j)} = \mathcal{T}_{\boldsymbol{\theta}}(\boldsymbol{s}, \boldsymbol{\xi}_j^{(0)}, \boldsymbol{e}_j)} \nabla_{\boldsymbol{\theta}} \mathcal{T}_{\boldsymbol{\theta}}(\boldsymbol{s}, \boldsymbol{\xi}_j^{(0)}, \boldsymbol{e}_j) \right\}, \quad (16)$$

where with a slight abuse of notation, we denote $\boldsymbol{\xi}_j^{(\ell)} = \boldsymbol{\xi}^{(\ell)}$ when $\ell > 0$.

We follow Haarnoja et al. (2018) to adaptively adjust the reward scaling coefficient $\alpha$. Denote $H_{\text{target}}$ as a fixed target entropy, heuristically chosen as $H_{\text{target}} = -\dim(\mathcal{A})$. We update $\alpha$ by performing gradient descent on $\eta := \log(\alpha)$ under the loss

$$J(\eta) = \mathbb{E}_{\boldsymbol{s} \sim \rho(\boldsymbol{s})}[\eta(-\log \pi_{\boldsymbol{\theta}}(\boldsymbol{a} \mid \boldsymbol{s}) - H_{\text{target}})], \quad (17)$$

where the marginal log-likelihood is estimated by $\log \pi_{\boldsymbol{\theta}}(\boldsymbol{a} \mid \boldsymbol{s}) = \log \frac{\sum_{\ell=0}^{L} \pi_{\boldsymbol{\theta}}(\boldsymbol{a} \mid \boldsymbol{s}, \boldsymbol{\xi}^{(\ell)})}{L+1}$, where $\boldsymbol{a} \sim \pi_{\boldsymbol{\theta}}(\cdot \mid \boldsymbol{s}, \boldsymbol{\xi}^{(0)})$ and $\boldsymbol{\xi}^{(0)}, \boldsymbol{\xi}^{(1)}, \ldots, \boldsymbol{\xi}^{(L)} \overset{iid}{\sim} p(\boldsymbol{\xi})$.

## 2.6 Off policy learning with IDAC

We incorporate the proposed twin-delayed DGNs and SIA into the off-policy framework. Specifically, the samples are gathered with a SIA based behavior policy and stored in a replay buffer. For each state $\boldsymbol{s}_t$, the agent will first sample a random noise $\boldsymbol{\xi}_t \sim p(\boldsymbol{\xi})$, then generate an action by $\boldsymbol{a}_t \sim \pi_{\boldsymbol{\theta}}(\boldsymbol{a}_t \mid \boldsymbol{s}_t, \boldsymbol{\xi}_t)$, and observe a reward $r_t$ and next state $\boldsymbol{s}_{t+1}$ returned by the environment. We save the tuples $(\boldsymbol{s}_t, \boldsymbol{a}_t, r_t, \boldsymbol{s}_{t+1})$ in a replay buffer and sample them uniformly when training the DGNs based implicit distributional critics and the SIA based semi-implicit policy. We provide an overview of the algorithm here and defer a pseudo code with all implementation details to Appendix B.

# 3 Experiments

Our experiments serve to answer the following questions: **(a)** How does IDAC perform when compared to state-of-the-art baselines, including SAC (Haarnoja et al., 2018), TD3 (Fujimoto et al., 2018), and PPO (Schulman et al., 2017)? **(b)** Can a semi-implicit policy capture complex distributional properties such as skewness, multi-modality, and covariance structure? **(c)** How well is the distributional matching when minimizing the quantile regression Huber loss? **(d)** How important is the type of policy distribution, such as a semi-implicit policy, a diagonal Gaussian policy, or a deterministic policy under this framework? **(e)** How much improvement does using distributional critics bring? **(f)** How critical is the twin-delayed network? **(g)** Will other baselines (such as SAC) benefit from using multiple actions ($J > 1$) for policy gradient estimation?

We will show two sets of experiments, one for **evaluation study** and the other for **ablation study**, to answer the aforementioned questions. The evaluation study will be addressing questions **(a)**-**(c)** and ablation study will be addressing **(d)**-**(g)**. In addition, we would like to emphasize the importance of the interaction between SIA and DGNs by comparisons between IDAC and its variants excluding either SIA or DGNs; we defer the results to Fig. 4 (b) in the Appendix.

As shown in Engstrom et al. (2019), the code-level implementation of different RL algorithms can lead to significant differences in their empirical performances and hence a fair comparison needs to be run on the same codebase. Thus all compared algorithms are either from, or built upon the *stable baselines* codebase (`https://github.com/hill-a/stable-baselines`) of Hill et al. (2018) to minimize the potential gaps caused by the differences of code-level implementations.

IDAC is implemented with a uniform set of hyperparameters to guarantee fair comparisons. Specifically, we use three separate fully-connected multilayer perceptrons (MLPs), which all have two 256-unit hidden layers and ReLU nonlinearities, to define the proposed SIA and two DGNs, respectively. Both $p(\boldsymbol{\xi})$ and $p(\boldsymbol{\epsilon})$ are $\mathcal{N}(\mathbf{0}, \mathbf{I}_5)$ and such a random noise, concatenated with the state $\boldsymbol{s}$, will be used as the input of its corresponding network. Note a similar semi-implicit construction has also been successfully applied in Wang and Zhou (2020) to address the multi-armed bandit problem, where there is no dependence between different states. We fix for all experiments the number of noise $\boldsymbol{\xi}^{(\ell)}$ as $L = 21$. We set the number of equally-spaced quantiles (the same as the number of $\boldsymbol{\epsilon}^{(k)}$) as $K = 51$ and number of auxiliary actions as $J = 51$ by default. A more detailed parameter setting can be found in Appendix C. We conduct empirical comparisons on the benchmark tasks provided by OpenAI Gym (Brockman et al., 2016) and MuJoCo simulators (Todorov et al., 2012).

## 3.1 Evaluation study to answer questions (a)-(c)

**(a):** We compare IDAC with SAC, TD3, and PPO on challenging continuous control tasks; each task is evaluated across 4 random seeds and the evaluation is done per 2000 steps with 5 independent rollouts using the most recent policy (to evaluate IDAC, we first sample $\boldsymbol{\xi} \sim p(\boldsymbol{\xi})$ and then use the mean of $\pi_{\boldsymbol{\theta}}(\boldsymbol{a} \,|\, \boldsymbol{s}, \boldsymbol{\xi})$ as action output). As shown in Fig. 1, IDAC outperforms all baseline algorithms with a clear margin across almost all tasks. More detailed numerical comparisons can be found in Table 1. We also provide in the Appendix performance comparison with SDPG. For all baselines, we use their default hyperparameter settings from the original papers. Notice that $J$, $K$, and $L$ are hyperparameters to set, and making them too small might prevent IDAC from taking full advantage of its distributional settings and hence lead to clearly degraded performance for some tasks. In this paper, to balance the performance and computational complexity, we choose moderate values of $J = 51$, $K = 51$, and $L = 21$ for all evaluations.

**(b):** We also check how well is semi-implicit policy and whether it can capture complex distributional properties. We defer the empirical improvement that semi-implicit policy brings to the ablation study part and only show the flexible distribution it supports here. Specifically, we generate this plot by sampling $\boldsymbol{a}_i \sim \pi_{\boldsymbol{\theta}}(\cdot \,|\, \boldsymbol{s})$ for $i = 1, \ldots, 1000$, and use these 1000 random actions samples (where $\boldsymbol{\theta}$ is the policy parameters at $10^4$ timestep while the total training steps is $10^6$), generated given a state $\boldsymbol{s}$, to visualize the empirical joint distribution of two selected dimensions of the action, and the marginal distributions at both dimensions. We have examined the policy distributions from both early and late stages to validate the effectiveness of SIA. An example result from an early stage is illustrated in Fig. 2 and that from a late stage is deferred to Fig. 6 in Appendix F. As shown in the left two panels of Fig. 2, the semi-implicit policy is capable of capturing sknewness, multi-modality, and dependencies between different dimensions, none of which are captured by the diagonal Gaussian

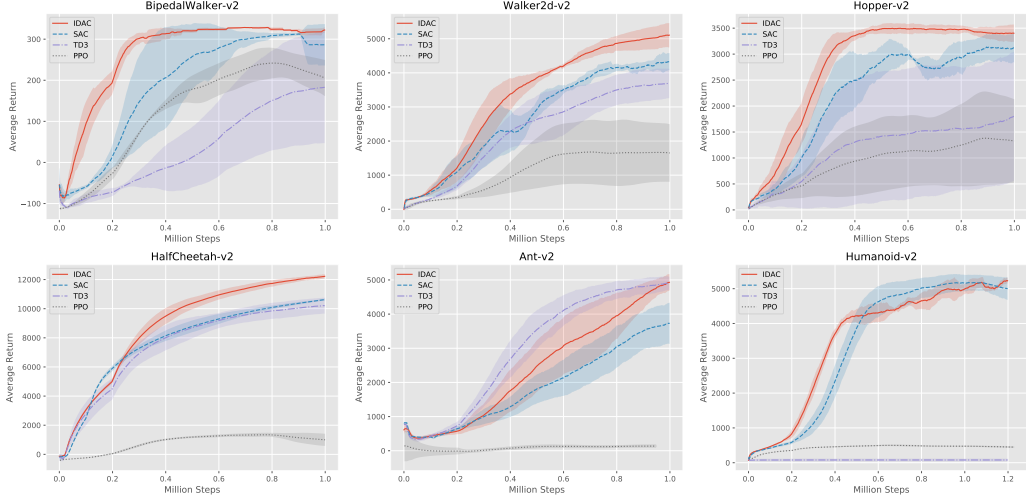

Figure 1: Training curves on continuous control benchmarks. The solid line is the average performance over 4 random seeds with ± 1 std shaded, and with a smoothing window of length 100.

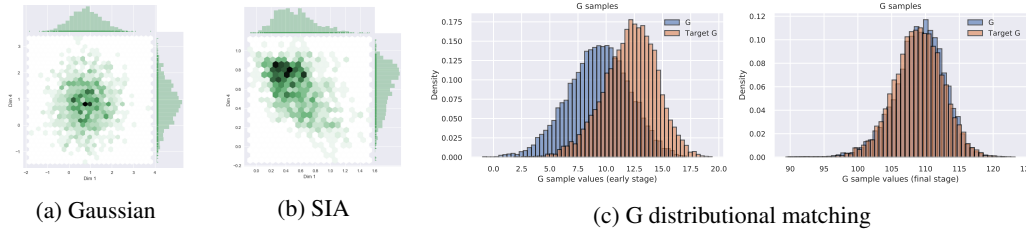

(a) Gaussian      (b) SIA           (c) G distributional matching

Figure 2: Visualization of Gaussian policy, SIA, and distributional matching for critic generators on Walker2d-v2 under SIA. Panels (a) and (b) show the density contour of 1000 randomly sampled actions at an early training stage, where x- and y-axis correspond to dimensions 1 and 4, respectively; Panel (c) shows the empirical density of 10000 DGN samples at an early training stage and the final one, where (target) $G$ samples are in (red) blue.

Table 1: Comparison of max average returns ± 1 std over 4 different random seeds.

|  | BipedalWalker | Walker2d | Hopper | HalfCheetah | Ant | Humanoid |
|---|---|---|---|---|---|---|
| PPO | $241.79 \pm 36.7$ | $1679.39 \pm 942.49$ | $1380.68 \pm 899.70$ | $1350.37 \pm 128.79$ | $141.79 \pm 451.10$ | $498.88 \pm 20.10$ |
| TD3 | $182.80 \pm 135.76$ | $3689.48 \pm 434.03$ | $1799.78 \pm 1242.63$ | $10209.65 \pm 548.14$ | $4905.74 \pm 203.09$ | $105.76 \pm 53.65$ |
| SAC | $312.48 \pm 2.81$ | $4328.95 \pm 249.27$ | $3138.93 \pm 299.62$ | $10626.34 \pm 73.78$ | $3732.23 \pm 602.83$ | $5055.64 \pm 62.96$ |
| IDAC | $\mathbf{328.44 \pm 1.23}$ | $\mathbf{5107.07 \pm 351.37}$ | $\mathbf{3497.86 \pm 93.30}$ | $\mathbf{12222.80 \pm 157.15}$ | $\mathbf{4930.73 \pm 242.78}$ | $\mathbf{5233.43 \pm 85.87}$ |

policy. Moreover, with both a normality test on the marginal of each dimension of the SIA policy and the Pearson correlation test on many randomly selected action dimension pairs, we verify that the SIA policy is significantly non-normal in its univariate marginals and captures the correlations between different action dimensions; see Appendix F for more details.

This flexible policy of SIA can be beneficial to exploration especially during the early training stages. Furthermore, capturing the correlation between action dimensions intuitively will lead to a better policy, $e.g.$, a robot learning to move needs to coordinate the movements of different legs.

(c): Similar to Singh et al. (2020), we check the matching situation of minimizing the quantile regression Huber loss. In detail, we generate 10000 random noises $\epsilon_k, \epsilon'_k \sim p(\epsilon)$ to obtain $\{G_{\tilde{\omega}_1}(s, a, \epsilon_k)\}_{k=1}^{10000}$ and $\{r(s, a) + \gamma \tilde{G}_{\tilde{\omega}_1}(s', a', \epsilon'_k)\}_{k=1}^{10000}$, and then compare their histograms to check if the empirical distributions are similar to each other. We list the distributions on both early and late stages to demonstrate the evolvement of the DGN. On an early stage, both the magnitude and shape of two distributions are very different, while their differences diminish at a fast pace along with the training process. It illustrates that the DGN is able to represent the distribution well defined by the distributional Bellman equation.

Table 2: Variants for ablation study.

| Ablations | IDAC | SAC | SAC-J[2] | SDPG[3] | SDPG-Twin | IDAC-Gaussian | IDAC-Implicit | IDAC-Single |
|---|---|---|---|---|---|---|---|---|
| Policy dist. | semi-implicit | Gaussian | Gaussian | deterministic | deterministic | Gaussian | implicit | semi-implicit |
| Distributional | yes | no | no | yes | yes | yes | yes | yes |
| Twin or single | twin | twin | twin | single | twin | twin | twin | single |

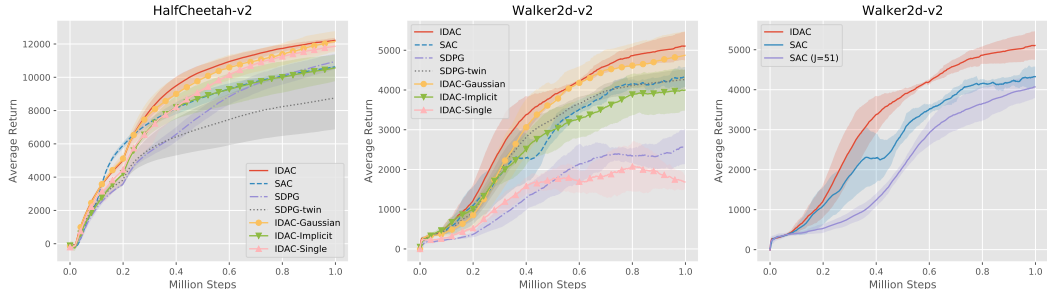

Figure 3: Training curves of ablation study.

## 3.2 Ablation study to answer questions (d)-(g)

We run a comprehensive set of ablation study to demonstrate the effectiveness of the SIA and DGNs on enhancing the performance. In general, there are three parts that we can control to see the differences they contribute: **(i)**: policy distribution {deterministic policy, Gaussian policy, semi-implicit policy, implicit policy}; **(ii)**: distributional aspect {no: action-value function, yes: distributional critic generator}; **(iii)**: prevent overestimation bias trick {single-delayed network, twin-delayed network}. Among those possible combinations, we choose a representative subset of them to show that the structure of IDAC is sound and brings notable improvement. They also answer the questions **(d)**-**(g)** outlined in the beginning. An implicit policy is constructed by deterministically transforming the concatenation of a random noise vector with state. In this way, the policy itself is still stochastic, but the log-likelihood is intractable and hence is not amenable to entropy regularization. We list all the 8 representative variants in Table 2, and evaluate their performances on HalfCheetah, Walker2d, and Ant environments with the same evaluation process described in Section 3.1.

**(d):** We make comparisons between IDAC, SDPG-Twin, IDAC-Gaussian, and IDAC-Implicit to demonstrate the superiority of using a semi-implicit policy. As shown in Fig. 3, we have IDAC > IDAC-Gaussian > SDPG-Twin > IDAC-Implicit, which not only demonstrates the improvement from the semi-implicit policy, but also implies the importance of using a stochastic policy with entropy regularization as shown in Haarnoja et al. (2018).

**(e):** The effect of the DGNs can be directly observed by comparing between IDAC-Gaussian and SAC, where IDAC-Gaussian is better than SAC on both tasks as shown in Fig. 3.

**(f):** To understand the importance of twin-delayed network structure, we make comparisons between SDPG with SDPG-Twin, and IDAC-Single with IDAC. As shown in Fig. 3, the one with the twin structure significantly outperform its counterpart without the twin structure in both cases, which demonstrate the effectiveness of the twin-delayed networks.

**(g):** Eventually, we want to demonstrate that the improvement of IDAC is not simply by sampling multiple actions for objective function estimation. As shown in the right panel of Fig. 3, the implementation of multiple actions on SAC does not boost the performance of SAC.

## 4 Conclusion

In this paper, we present implicit distributional actor-critic (IDAC), an off-policy based actor-critic algorithm incorporated with distributional learning. We model the return distribution with a deep generative network (DGN) and the policy with a semi-implicit actor (SIA), and mitigate the overestimation issue with a twin-delayed DGNs structure. We validate the critical roles of these components with a detailed ablation study, and demonstrate that IDAC is capable of the state-of-the-art performance on a number of challenging continuous control problems.

## Broader Impact

This work proposes a high sample-efficient algorithm to solve challenging continuous control reinforcement learning (RL) problems. RL could be applied to a wide range of applications, including resources management (Mao et al., 2016), traffic light control (Arel et al., 2010), optimizing chemistry reactions (Zhou et al., 2017), to name a few. Under the RL framework, sample efficiency is one of the top concerns for practicality, because the interactions between the agent and environment in reality is costly and the algorithm will be impractical if the number of interactions is demanding. Though there have been successfully cases of implementing RL algorithms to solve real-life problems, most of them need to be simulatable in nature to meet the demand of enough samples. Fortunately, lots of great works have focused on overcoming this challenge, and significant improvement has been made.

Ever since the birth of $Q$-learning algorithm in 1992 (Watkins and Dayan, 1992), the functionality of RL algorithms has grow rapidly. When $Q$-learning is first proposed, it can only be applied to toy examples such as "route finding" problems, and now it can even be applied to play Go (David et al., 2017) and defeat top human players. Moreover, with the help of deep learning, RL algorithms have shown promising performances on complicated computer games such as Dota (OpenAI, 2018) and StarCraft (Vinyals et al., 2019). All this accomplishments cannot be achieved without the consecutive effort put on improving the sample efficiency. In our new algorithm IDAC, we incorporate the advanced distributional idea with the off-policy stochastic policy setting, and obtain notable improvement over a number of state-of-the-art algorithms. This result is very promising and has huge potential to be applied or further improved to facilitate RL algorithm being implemented in more complicated real-life tasks such as self-driving cars and automation. However, such implementations need taking special care because it involves human beings, and the risk sensitivity is not part of the research of our work. We encourage further research taking risk into account so that it can be combined with IDAC to be applicable to a broader range of settings.

## Acknowledgements

The authors thank Mauricio Tec for many useful discussions. The authors thank the anonymous Area Chair for engaging the anonymous reviewers in the discussion and communicating with the authors through the CMT3 conference submission system to ask for the clarification of a remaining concern, and the anonymous reviewers for their valuable comments and suggestions that have helped us to improve the paper. The authors acknowledge the support of Grants IIS-1812699 and ECCS-1952193 from the U.S. National Science Foundation, the APX 2019 project sponsored by the Office of the Vice President for Research at The University of Texas at Austin, the support of NVIDIA Corporation with the donation of the Titan Xp GPU used for this research, and the Texas Advanced Computing Center (TACC) at The University of Texas at Austin for providing HPC resources that have contributed to the research results reported within this paper. URL: `http://www.tacc.utexas.edu`

## Footnotes

[2]SAC-J refers to SAC with $J$ actions to estimate its objective function.

[3]Note that the SDPG paper (Singh et al., 2020) is using a different codebase; the implementation-level differences make their reported results not directly comparable; we use this variant to illustrate how each component works.

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
