[Supplementary Material]

# Implicit Distributional Reinforcement Learning: Appendix

## A   Proof of Lemma 1

Denote

$$\mathcal{H} = \mathbb{E}_{\boldsymbol{a}\sim\pi_{\boldsymbol{\theta}}(\boldsymbol{a}|\boldsymbol{s})}\log\pi_{\boldsymbol{\theta}}(\boldsymbol{a}|\boldsymbol{s}),$$

and

$$\mathcal{H}_L = \mathbb{E}_{\boldsymbol{\xi}^{(0)},...,^{(L)}\sim p(\boldsymbol{\xi})}\mathbb{E}_{\boldsymbol{a}\sim\pi_{\boldsymbol{\theta}}(\boldsymbol{a}\,|\,\boldsymbol{s},\boldsymbol{\xi}^{(0)})}\log\frac{1}{L+1}\sum_{\ell=0}^{L}\pi_{\boldsymbol{\theta}}(\boldsymbol{a}\,|\,\boldsymbol{s},\boldsymbol{\xi}^{(\ell)}),$$

and

$$\pi_{\boldsymbol{\theta}}(\boldsymbol{a}|\boldsymbol{s},\boldsymbol{\xi}^{(0):(L)}) = \frac{1}{L+1}\sum_{\ell=0}^{L}\pi_{\boldsymbol{\theta}}(\boldsymbol{a}\,|\,\boldsymbol{s},\boldsymbol{\xi}^{(\ell)}).$$

Notice that $\boldsymbol{\xi}$s are from the same distribution, so we have

$$\mathcal{H}_L = \frac{1}{L+1}\sum_{i=0}^{L}\mathbb{E}_{\boldsymbol{\xi}^{(0)},...,^{(L)}\sim p(\boldsymbol{\xi})}\mathbb{E}_{\boldsymbol{a}\sim\pi_{\boldsymbol{\theta}}(\boldsymbol{a}\,|\,\boldsymbol{s},\boldsymbol{\xi}^{(i)})}\log\frac{1}{L+1}\sum_{\ell=0}^{L}\pi_{\boldsymbol{\theta}}(\boldsymbol{a}\,|\,\boldsymbol{s},\boldsymbol{\xi}^{(\ell)})$$

$$= \mathbb{E}_{\boldsymbol{\xi}^{(0)},...,^{(L)}\sim p(\boldsymbol{\xi})}\mathbb{E}_{\boldsymbol{a}\sim\pi_{\boldsymbol{\theta}}(\boldsymbol{a}\,|\,\boldsymbol{s},\boldsymbol{\xi}^{(0):(L)})}\log\pi_{\boldsymbol{\theta}}(\boldsymbol{a}|\boldsymbol{s},\boldsymbol{\xi}^{(0):(L)}).$$

Use the identity that $\mathbb{E}_{\boldsymbol{a}\sim\pi_{\boldsymbol{\theta}}(\boldsymbol{a}|\boldsymbol{s})} = \mathbb{E}_{\boldsymbol{\xi}^{(0)},...,^{(L)}\sim p(\boldsymbol{\xi})}\mathbb{E}_{\boldsymbol{a}\sim\pi_{\boldsymbol{\theta}}(\boldsymbol{a}\,|\,\boldsymbol{s},\boldsymbol{\xi}^{(0):(L)})}$, we can rewrite $\mathcal{H}$ as

$$\mathcal{H} = \mathbb{E}_{\boldsymbol{\xi}^{(0)},...,^{(L)}\sim p(\boldsymbol{\xi})}\mathbb{E}_{\boldsymbol{a}\sim\pi_{\boldsymbol{\theta}}(\boldsymbol{a}\,|\,\boldsymbol{s},\boldsymbol{\xi}^{(0):(L)})}\log\pi_{\boldsymbol{\theta}}(\boldsymbol{a}|\boldsymbol{s}).$$

Therefore, we have

$$\mathcal{H}_L - \mathcal{H} = \mathbb{E}_{\boldsymbol{\xi}^{(0)},...,^{(L)}\sim p(\boldsymbol{\xi})}\mathbb{E}_{\boldsymbol{a}\sim\pi_{\boldsymbol{\theta}}(\boldsymbol{a}\,|\,\boldsymbol{s},\boldsymbol{\xi}^{(0):(L)})}\log\frac{\pi_{\boldsymbol{\theta}}(\boldsymbol{a}|\boldsymbol{s},\boldsymbol{\xi}^{(0):(L)})}{\pi_{\boldsymbol{\theta}}(\boldsymbol{a}|\boldsymbol{s})}$$

$$= \mathrm{KL}(\pi_{\boldsymbol{\theta}}(\boldsymbol{a}|\boldsymbol{s},\boldsymbol{\xi}^{(0):(L)})||\pi_{\boldsymbol{\theta}}(\boldsymbol{a}|\boldsymbol{s})) \geq 0.$$

To compare between $\mathcal{H}_L$ and $\mathcal{H}_{L+1}$, rewrite $\mathcal{H}_L$ as

$$\mathcal{H}_L = \mathbb{E}_{\boldsymbol{\xi}^{(0)},...,^{(L)},^{(L+1)}\sim p(\boldsymbol{\xi})}\mathbb{E}_{\boldsymbol{a}\sim\pi_{\boldsymbol{\theta}}(\boldsymbol{a}\,|\,\boldsymbol{s},\boldsymbol{\xi}^{(0):(L)})}\log\pi_{\boldsymbol{\theta}}(\boldsymbol{a}|\boldsymbol{s},\boldsymbol{\xi}^{(0):(L)})$$

and $\mathcal{H}_{L+1}$ as

$$\mathcal{H}_{L+1} = \mathbb{E}_{\boldsymbol{\xi}^{(0)},...,^{(L)},^{(L+1)}\sim p(\boldsymbol{\xi})}\mathbb{E}_{\boldsymbol{a}\sim\pi_{\boldsymbol{\theta}}(\boldsymbol{a}\,|\,\boldsymbol{s},\boldsymbol{\xi}^{(0)})}\log\pi_{\boldsymbol{\theta}}(\boldsymbol{a}|\boldsymbol{s},\boldsymbol{\xi}^{(0):(L+1)})$$

$$= \mathbb{E}_{\boldsymbol{\xi}^{(0)},...,^{(L)},^{(L+1)}\sim p(\boldsymbol{\xi})}\mathbb{E}_{\boldsymbol{a}\sim\pi_{\boldsymbol{\theta}}(\boldsymbol{a}\,|\,\boldsymbol{s},\boldsymbol{\xi}^{(0):(L)})}\log\pi_{\boldsymbol{\theta}}(\boldsymbol{a}|\boldsymbol{s},\boldsymbol{\xi}^{(0):(L+1)})$$

and the difference will be

$$\mathcal{H}_L - \mathcal{H}_{L+1} = \mathbb{E}_{\boldsymbol{\xi}^{(0)},...,^{(L)},^{(L+1)}\sim p(\boldsymbol{\xi})}\mathbb{E}_{\boldsymbol{a}\sim\pi_{\boldsymbol{\theta}}(\boldsymbol{a}\,|\,\boldsymbol{s},\boldsymbol{\xi}^{(0):(L)})}\big[\log\pi_{\boldsymbol{\theta}}(\boldsymbol{a}|\boldsymbol{s},\boldsymbol{\xi}^{(0):(L)}) - \log\pi_{\boldsymbol{\theta}}(\boldsymbol{a}|\boldsymbol{s},\boldsymbol{\xi}^{(0):(L+1)})\big]$$

$$= \mathbb{E}_{\boldsymbol{\xi}^{(0)},...,^{(L)},^{(L+1)}\sim p(\boldsymbol{\xi})}\mathrm{KL}(\pi_{\boldsymbol{\theta}}(\boldsymbol{a}|\boldsymbol{s},\boldsymbol{\xi}^{(0):(L)})||\pi_{\boldsymbol{\theta}}(\boldsymbol{a}|\boldsymbol{s},\boldsymbol{\xi}^{(0):(L+1)})) \geq 0.$$

Finally, we arrive at the conclusion that for any $\ell$, we have

$$\mathcal{H} \leq \mathcal{H}_{\ell+1} \leq \mathcal{H}_\ell.$$

# B   Detailed pseudo code

---

**Algorithm 2** Implicit Distributional Actor-Critic (IDAC)

---

**Require:** Learning rate $\lambda$, batch size $M$, quantile number $K$, action number $J$ and noise number $L$, target entropy $\mathcal{H}_t$.

Initial policy network parameter $\boldsymbol{\theta}$, action-value function network parameter $\boldsymbol{\omega}_1, \boldsymbol{\omega}_2$, entropy parameter $\eta$.

Initial target network parameter $\tilde{\boldsymbol{\omega}}_1 = \boldsymbol{\omega}_1, \tilde{\boldsymbol{\omega}}_2 = \boldsymbol{\omega}_2$.

**for** the number of environment steps **do**

Sample $M$ number of transitions $\{\boldsymbol{s}_t^i, \boldsymbol{a}_t^i, r_t^i, \boldsymbol{s}_{t+1}^i\}_{i=1}^M$ from the replay buffer

Sample $\boldsymbol{\epsilon}_t^{i,(k)}, \boldsymbol{\epsilon}_{t+1}^{i,(k)}, \boldsymbol{\xi}_{t+1}^{i,(\ell)}$ from $\mathcal{N}(\boldsymbol{0}, \mathbf{I})$ for $i = 1 \cdots M$ and $k = 1 \cdots K$ and $\ell = 0 \cdots L$.

Sample $\boldsymbol{a}_{t+1}^i \sim \pi_{\boldsymbol{\theta}}(\cdot \mid \boldsymbol{s}_{t+1}^i, \boldsymbol{\xi}_{t+1}^{i,(0)}) = \mathcal{N}(\mathcal{T}_{\boldsymbol{\theta}}^1(\boldsymbol{s}_{t+1}^i, \boldsymbol{\xi}_{t+1}^{i,(0)}), \mathcal{T}_{\boldsymbol{\theta}}^2(\boldsymbol{s}_{t+1}^i, \boldsymbol{\xi}_{t+1}^{i,(0)}))$ for $i = 1 \cdots M$.

Apply Bellman update to create samples (of return distribution)

$$y_{1,i,k} = r_t^i + \gamma G_{\tilde{\boldsymbol{\omega}}_1}(\boldsymbol{s}_{t+1}^i, \boldsymbol{a}_{t+1}^i, \boldsymbol{\epsilon}_{t+1}^{i,(k)}) \quad \text{\# Calculate target values}$$

$$y_{2,i,k} = r_t^i + \gamma G_{\tilde{\boldsymbol{\omega}}_2}(\boldsymbol{s}_{t+1}^i, \boldsymbol{a}_{t+1}^i, \boldsymbol{\epsilon}_{t+1}^{i,(k)}) \quad \text{\# Calculate target values}$$

and let

$$(\overrightarrow{y}_{1,i,1}, \ldots, \overrightarrow{y}_{1,i,K}) = \text{StopGradient}(\text{sort}(y_{1,i,1}, \ldots, y_{1,i,K})) \quad \text{\# Obtain target quantile estimation}$$

$$(\overrightarrow{y}_{2,i,1}, \ldots, \overrightarrow{y}_{2,i,K}) = \text{StopGradient}(\text{sort}(y_{2,i,1}, \ldots, y_{2,i,K})) \quad \text{\# Obtain target quantile estimation}$$

$$\overrightarrow{y}_{i,k} = \min(\overrightarrow{y}_{1,i,k}, \overrightarrow{y}_{2,i,k}), \text{ for } i = 1 \cdots M; k = 1 \cdots K$$

Generate samples $x_{1,i,k} = G_{\boldsymbol{\omega}_1}(\boldsymbol{s}_t^i, \boldsymbol{a}_t^i, \boldsymbol{\epsilon}_t^{i,(k)})$ and $x_{2,i,k} = G_{\boldsymbol{\omega}_2}(\boldsymbol{s}_t^i, \boldsymbol{a}_t^i, \boldsymbol{\epsilon}_t^{i,(k)})$, and let

$$(\overrightarrow{x}_{1,i,1}, \ldots, \overrightarrow{x}_{1,i,K}) = \text{sort}(x_{1,i,1}, \ldots, x_{1,i,K})$$

$$(\overrightarrow{x}_{2,i,1}, \ldots, \overrightarrow{x}_{2,i,K}) = \text{sort}(x_{2,i,1}, \ldots, x_{2,i,K})$$

Update action-value function parameter $\boldsymbol{\omega}_1$ and $\boldsymbol{\omega}_2$ by minimizing the quantile loss

$$J(\boldsymbol{\omega}_1, \boldsymbol{\omega}_2) = \frac{1}{M} \sum_{i=1}^M \frac{1}{K^2} \sum_{k=1}^K \sum_{k'=1}^K \rho_{\tau_k}^\kappa(\overrightarrow{y}_{i,k} - \overrightarrow{x}_{1,i,k'}) + \frac{1}{M} \sum_{i=1}^M \frac{1}{K^2} \sum_{k=1}^K \sum_{k'=1}^K \rho_{\tau_k}^\kappa(\overrightarrow{y}_{i,k} - \overrightarrow{x}_{2,i,k'}).$$

Sample $\Xi_t^{i,h}, \boldsymbol{\epsilon}_t^{i,(j)}$ from $\mathcal{N}(\boldsymbol{0}, \mathbf{I})$, for $i = 1 \cdots M, j = 1 \cdots J$ and $h = 0 \cdots L + J$, and form $\boldsymbol{\xi}_t^{i,(j,\ell)}$ from $\Xi_t^{i,h}$ by concatenating $L$ of them to the rest of $J$s. Sample $\boldsymbol{a}_t^{i,(j)} \sim \pi_{\boldsymbol{\theta}}(\cdot \mid \boldsymbol{s}_t^i, \boldsymbol{\xi}_t^{i,(j,0)}) = \mathcal{N}(\mathcal{T}_{\boldsymbol{\theta}}^1(\boldsymbol{s}_t^i, \boldsymbol{\xi}_t^{i,(j,0)}), \mathcal{T}_{\boldsymbol{\theta}}^2(\boldsymbol{s}_t^i, \boldsymbol{\xi}_t^{i,(j,0)}))$ using

$$\boldsymbol{a}_t^{i,(j)} = \mathcal{T}_{\boldsymbol{\theta}}(\boldsymbol{s}_t^i, \boldsymbol{\xi}_t^{i,(j,0)}, \boldsymbol{e}_t^i) = \mathcal{T}_{\boldsymbol{\theta}}^1(\boldsymbol{s}_t^i, \boldsymbol{\xi}_t^{i,(j,0)}) + \boldsymbol{e}_t^i \odot \mathcal{T}_{\boldsymbol{\theta}}^2(\boldsymbol{s}_t^i, \boldsymbol{\xi}_t^{i,(j,0)}), \ \boldsymbol{e}_t^i \sim \mathcal{N}(\boldsymbol{0}, \mathbf{I})$$

for $i = 1, \cdots, M$.

Update the policy function parameter $\boldsymbol{\theta}$ by minimizing

$$J(\boldsymbol{\theta}) = -\frac{1}{M} \sum_{i=1}^M \left\{ \frac{1}{2J} \sum_{z=1}^2 \sum_{j=1}^J G_{\boldsymbol{\omega}_z}(\boldsymbol{s}_t^i, \boldsymbol{a}_t^{i,(j)}, \boldsymbol{\epsilon}_t^{i,(j)}) - \exp(\eta) \sum_{j=1}^J \frac{1}{J} \left[ \log \frac{\sum_{\ell=0}^L \pi_{\boldsymbol{\theta}}(\boldsymbol{a}_t^{i,(j)} \mid \boldsymbol{s}_t^i, \boldsymbol{\xi}_t^{i,(j,\ell)})}{L+1} \right] \right\}.$$

We also use stop gradient on $(\mathcal{T}_{\boldsymbol{\theta}}^1(\boldsymbol{s}_t^i, \boldsymbol{\xi}_t^{i,(j)}), \mathcal{T}_{\boldsymbol{\theta}}^2(\boldsymbol{s}_t^i, \boldsymbol{\xi}_t^{i,(j)}))$ to reduce variance on gradient as mentioned in Eq (16).

Update the log entropy parameter $\eta$ by minimizing

$$J(\eta) = \frac{1}{M} \sum_{i=1}^M [\text{StopGradient}(-\log \frac{\sum_{\ell=0}^L \pi_{\boldsymbol{\theta}}(\boldsymbol{a}_t^{i,(0)} \mid \boldsymbol{s}_t^i, \boldsymbol{\xi}_t^{i,(\ell)})}{L+1} - \mathcal{H}_t)\eta]$$

**end for**

---

# C  Hyperparameters of IDAC

Table 3: IDAC hyperparameters

| Parameter | Value |
|---|---|
| Optimizer | Adam |
| learning rate | 3e-4 |
| discount | 0.99 |
| replay buffer size | $10^6$ |
| number of hidden layers (all networks) | 2 |
| number of hidden units per layer | 256 |
| number of samples per minibatch | 256 |
| entropy target | $-\dim(\mathcal{A})$ (*e.g.*, $-6$ for HalfCheetah-v2) |
| nonlinearity | ReLU |
| target smoothing coefficient | 0.005 |
| target update interval | 1 |
| gradient steps | 1 |
| distribution of $\boldsymbol{\xi}$ | $\mathcal{N}(\mathbf{0}, \mathbf{I}_5)$ |
| distribution of $\boldsymbol{\epsilon}$ | $\mathcal{N}(\mathbf{0}, \mathbf{I}_5)$ |
| J | 51 |
| K | 51 |
| L | 21 |

# D  Additional ablation study

Additional ablation studies on Ant is shown in Fig. 4a for a thorough comparison. In Ant, the performance of IDAC is on par with that of IDAC-Gaussian, which outperforms the other variants.

Furthermore, we would like to learn the interaction between DGN and SIA by running ablation studies by holding each of them as a control factor; we conduct the corresponding experiments on Walker2d. From Fig. 4b, we can observe that by removing either SIA (resulting in IDAC-Gaussian) or D$G$N (resulting in IDAC-noD$G$N) from IDAC in general negatively impacts its performance, which echoes our motivation that we integrate D$G$N and SIA to allow them to help strengthen each other: *(i)* Modeling $G$ exploits distributional information to help better estimate its mean $Q$ (note C51, which outperforms D$Q$N by exploiting distributional information, also conducts its argmax operation on $Q$); *(ii)* A more flexible policy may become more necessary given a better estimated $Q$.

(a) Additional ablation study on Ant    (b) Additional ablation study on Walker2d

Figure 4: Additional plots for ablation study

# E    Additional comparison with SDPG

In Fig. 5, we include a thorough comparison with SDPG (implemented based on the *stable baselines* codebase).

Figure 5: Training curves on continuous control benchmarks. The solid line is the average performance over 4 seeds with $\pm$ 1 std shaded, and with a smoothing window of length 100.

# F    Late stage policy visualization

We show in Fig. 6 the visualization of the late stage policy of one seed from Walker2d-v2 environment. We can see that SIA does provide a more flexible policy even in the late stage, where the correlations between action dimensions is clear on plot, and the marginal distributions are more flexible than a Gaussian distribution. Moreover, we randomly choose 1000 states and conduct normality tests as well as correlation tests on them. As a result, **all** of the tests indicate that the SIA policy captures the non-zero correlations between the selected dimensions, and the marginal distributions for each dimension across different states are significantly non-normal.

Figure 6: Visualization of the SIA on Walker2d-v2. The density contour of 1000 randomly sampled actions at a late training stage, where the x- and y-axis correspond to dimensions 1 and 4, respectively.