[Reviews · NeurIPS 2020]

Review 1

Summary and Contributions: This paper introduces an extension to SAC/TD3 type actor-critic algorithms. It models the critic with a generator network (actually twin delayed networks) and models the stochastic actor as a diagonal gaussian which also receives a noise input. The paper includes derivations required for approximating the entropy of their semi-implicit actor along with experiments showcasing improved performance on a standard sweet of deep RL control problems.

Strengths: This work provides a good balance between analytical and empirical results. The method appears to be sound and does introduce a novel component--the semi-implicit actor and associated entropy estimation. The empirical results show a clear trend of improved performance over previous algorithms on an appropriate suite of RL tasks. And their ablation experiments are helpful at establishing the importance of each component of their overall algorithm.

Weaknesses: One possible weakness of this work is the somewhat incremental nature of the improvement. For example one of the stated contributions is to apply the twin-delayed technique to learning the critic, which is well established.

Correctness: Yes, the algorithm and derivation of loss functions appear correct, and then empirical methodology appears to compare the new algorithm against baseline techniques on a fair basis, with any possible confounding factors removed. Since the Ant-v2 environment is one in which the TD3 baseline outperforms the proposed method, it would be interesting to include ablation studies on this environment, as well. And likewise with Humanoid, since it has such a large performance gap between SAC and TD3. The authors claim that the semi-implicit actor can represent more complicated policy distributions, include skewness, etc., even through the diagonal-gaussian parameterization, and that this capability is useful for learning more complicated tasks. This claim is straightforwardly appealing, and is supported by their ablation experiments (Fig 3) and the visualization of the policy (Fig 2b). However the claim could be made much more strongly, for example by tying the distribution in Fig 2b to a particular behavior pattern which was not possible with the diagonal gaussian alone, or by showing an example like in Fig 2b but from later in training. As written, Fig 2b is from early in training, which could mean its very non-diagonal shape is more a result of random initialization than of learning?

Clarity: The paper is mostly well organized and written. Overall I would encourage some minor revisions for clarity in due course.

Relation to Prior Work: Most of the discussion on prior work is satisfactory, however there appears to be a lot of overlap between the methods/developments in this paper and in the paper cited for the algorithm SDPG. It would be helpful to draw more explicit parallels/distinctions from that work, otherwise it appears at first glance to make the current work more incremental. Overall the related work section could be expanded slightly to discuss more explicitly the implications from prior work on the current paper, slightly beyond the brief statements of existence currently included.

Reproducibility: Yes

Additional Feedback:


Review 2

Summary and Contributions: The paper considers the model-free Reinforcement Learning setting with continuous actions, and proposes a new algorithm that increases sample-efficiency. The proposed algorithm consists of two elements: a novel critic, built on ideas from the Distributional RL literature, but adapted to continuous actions, and a novel actor, that combines an explicit component (a Gaussian-based function) and an implicit component (one of the inputs of the Gaussian is a randomly-sampled epsilon, as in Generative Networks). The resulting algorithm is empirically shown to lead to state-of-the-art results.

Strengths: The paper is extremely well-written, clear and compact. The amount of equations in the paper illustrates how much information has to be compactly conveyed by the paper, yet everything is easy to follow (for an expert in discrete-action RL, but not continuous actions). The proposed algorithm is quite complicated, and has many original components to it, but every component is clearly motivated (the distributional aspect is motivated, as is why two critics and a minimum are used, or why a semi-implicit actor is necessary). The empirical evaluation is convincing, and the authors take care to evaluate the fairness of their evaluation, in their case by ensuring that all the algorithms are built on the same code-base. The results presented in the paper are highly significant, and the algorithm itself makes few assumptions (not more than PPO or other continuous-action algorithms). As such, it can be applied to a wide variety of problems. An added bonus is that the code of the proposed algorithm is available in the supplementary material, which ensures that a baseline exists, and that the paper can be reproduced.

Weaknesses: The compactness of the papers led me to miss the exact definition of the sorting operation happening in the critic (the authors addressed that concern by clarifying their notations in the author response, I suggest that they put that clarification in the paper). For reference, my remark was as follows: the critic algorithm is built around a sorting operation, that is not fully described on two levels. First, it is unclear whether the sorting function is applied to a single vector of floating-point values, and sorts them (as Equation 4 would indicate, with arrows put above vectors), or whether vectors in a batch of vectors are being sorted (as relatively explicitly shown between Equations 8 and 9). In the second case, how vectors are sorted should have been defined, as it is unclear how they are compared (are their norms being compared, or is a hierarchical comparison between their components being performed?). The presence of source code allows an answer for this question to be found relatively easily, but the absence of that little practical details makes imagining the algorithm as the paper is read more difficult.

Correctness: The theoretical contributions in the paper flow well, and do not raise questions about correctness. The empirical evaluation is thorough, fair, and uses modern, high-performance algorithms as baselines.

Clarity: The paper is well-written and easy to understand, even for someone familiar with most of the concepts presented in the paper (distributional RL, GAN networks, the Soft Actor-Critic), but not expert in these particular domains. Someone less familiar with the work the paper builds on may have a harder time following the paper, as it remains quite dry, but I don't see how the paper could be made easier on the reader without exceeding the page limit.

Relation to Prior Work: The paper thoroughly discusses related work and emphasizes its contributions in a satisfactory way. There are some cases, such as Equation 5, where it is a bit more difficult to see if the contribution of the paper is the use of an existing equation in a new setting, or a new equation. Sentences such as "Inspired from [X], we propose to use something that looks a bite like [Y]" often lead to confusion, as they indicate that the equation that will follow will not be X, but the sentence does not indicate whether Y contains X (and so, the following equation is Y in a new context), or if Y did not know about X, and the following equation is an original combination of X and Y.

Reproducibility: Yes

Additional Feedback: There was a discussion among the reviewers about the novelty of this paper, and its relation with other work on distributional RL and generative methods. The novelty of this paper became clear after very attentive reading. I therefore suggest that the novel components of the actor and critic presented in this paper are better emphasized in the introduction of the paper, and compared to other work.


Review 3

Summary and Contributions: This paper proposes a new algorithm (implicit distributional actor critic; IDAC) which uses a distributional critic and a richer representation of the actor policy (semi-implicit actor; SIA). The implicit distributional critic uses twin delayed deep generator networks to limit over-estimation problems, and the implicit policy is argued to provide improved exploration. Performance is evaluated, in comparison with SAC/TD3/PPO, on several Mujoco continuous control environments.

Strengths: I would consider the structure of the empirical work to be one of the highlights of the paper. Besides the generally positive results, the authors do a good job of proposing clear experimental questions and systematically evaluating them. The ablation study in particular helps to support the proposed setup in this work.

Weaknesses: Except for in ablation, the proposed method is not sufficiently compared against other similar approaches. That is, while it is compared with other RL algorithms, the only other method compared with that is using similar approaches for the critic or actor (both of which have closely related work) was an extremely recent unpublished work on SDPG, and this was only shown in the ablation study (as opposed to over the larger set of environments in Figure 1). My concern here is that while this paper shows that a distributional critic and implicit policy improve performance, it is not entirely clear that the proposed form of these is any better than those already proposed in prior work. Again, with the caveat that they do compare with a method from a very recent paper that has not been published, as well as that the ablation does (to a limited degree) cover some variations in the approach.

Correctness: One of the claims in the contributions was that the flexibility of SIA would improve exploration. This does not seem to have been evaluated, and while I share the author’s intuition on this, it is something that should be phrased as a hypothesis rather than a contribution. Section 2.2 seems to want a formal result showing that this approach to learning the distribution of returns is principled. As there is no consistent connection between \epsilon^{(k)} and \tau_i, I do not see why the proposed method is necessarily better than simply minimizing the sample Wasserstein, which as noted can produce biased gradients.

Clarity: I thought this paper was reasonably clear and well written.

Relation to Prior Work: Overall, I like the direction of the work, and agree with the authors point about the implicit parameterization of the actor potentially leading to better exploration. I think this would be a great contribution if it was properly investigated. I have some minor concerns about the convergence properties of the proposed distributional critic. Finally, my biggest issue is with regards to related work. There’s a good chance that the choices made in this paper are actually superior to those made in the paper I mentioned, but this work would need to have significant chances to allow a clear comparison with it. Pseudo-code in the appendix *really* stretches the boundary between mathematical definition and pseudo-code. Providing actual pseudo-code would be preferable. L128: “In the same sprite” perhaps should be “spirit”?

Reproducibility: Yes

Additional Feedback: Overall, I like the direction of the work, and agree with the authors point about the implicit parameterization of the actor potentially leading to better exploration. I think this would be a great contribution if it was properly investigated. I have some minor concerns about the convergence properties of the proposed distributional critic. Finally, my biggest issue is with regards to related work. There’s a good chance that the choices made in this paper are actually superior to those made in the paper I mentioned, but this work would need to have significant chances to allow a clear comparison with it. Pseudo-code in the appendix *really* stretches the boundary between mathematical definition and pseudo-code. Providing actual pseudo-code would be preferable. L128: “In the same sprite” perhaps should be “spirit”? -------------------------------------------------------- Update: After reading the rebuttal and discussing with other reviewers, I believe the author's have addressed my main concerns and have updated my score. I would emphasize the author(s) discussion of the differences with DPO should be included in the final version in some form.


Review 4

Summary and Contributions: This paper proposes to use more flexible parameterizations for distributional Q-learning and for continuous-action policies, aiming to better model the maximum-entropy policy distribution in a soft actor critic-like setting. It introduces (1) an implicit distributional value function, which produces a sampled value estimate given a state, action, and a noise vector; (2) a semi-implicit policy parameterization, which can represent richer distributions than the typical Gaussian policy; and (3) tractable learning algorithms for both value function and policy. The paper includes thorough experimental results on standard benchmarks, as well as ablations to attempt to show the contribution of each component of the method. **Post rebuttal** Thanks to the authors for your response, which cleared things up for me. I stand by my original score and think this would be a solid paper for NeurIPS.

Strengths: This work introduces a semi-parametric policy class which is interesting and seems to be able to capture more complex multi-modal structure. It provides evidence that richer policy parameterizations can lead to improved performance, although as has been observed in the past with mixture-of-Gaussians or normalizing flows policies this gain is typically small. The multimodal structure of the learned policy is verified empirically. The ideas introduced around training the implicit distributional value function are new to me, as is the elementwise minimization over the sampled values. The ablation study is thorough and may be of use to others in the community who are attempting to make decisions about which methods to use, e.g. implicit policy versus Gaussian policy or single critic versus twin-delayed critic.

Weaknesses: Some decisions in the paper are not well motivated, and despite the extensive set of ablations the importance of some choices remains unclear. There are really two separate methodological improvements proposed in this paper: the implicit distributional value function and the semi-implicit policy. These two components might have been better off proposed separately so that they could be studied in more detail. One paper could propose the implicit parameterization of the distributional value function and compare its results to C51 and QR-DQN, while another used a standard expected-value critic with the semi-implicit policy and evaluated in detail the impact of the policy parameterization compared to Gaussian, mixture of Gaussian, and normalizing flow policies. Further complicating matters, there are a lot of bells and whistles in the final method (twin delayed critics, learned temperature, etc). While ablations help a lot, all the tricks do muddle my understanding of which improvements are contributing what. I do not understand clearly the motivation behind using an implicit distribution for the value function. With a 1D function, especially given that the loss being used is based on quantiles anyway, what is to be gained by using an implicit distribution instead of quantile regression?

Correctness: I'm probably missing something, but where did the normalizing integral for G go between equations 13 and 14? Should the $\pi_\theta(a^{(1)} | s, \xi^{(\ell)})$ have $a^{(\ell)}$ instead? As I understand it the SAC results are using the original version of SAC [16] rather than the modern version [41] with the learned temperature — please clarify if this is incorrect. Given that this method uses the learned temperature parameter from [41], it should compare against that version of SAC.

Clarity: Overall the paper is acceptably well-written, but it could be clearer. Due to having two largely independent contributions in the same paper, things can get a bit muddled at times. In the caption for Table 1, what is meant by "average maximal returns"? What is being maximized over?

Relation to Prior Work: Yes

Reproducibility: Yes

Additional Feedback: It would really help the work to very precisely state in the introduction what the problem is that you're trying to solve, the method you're proposing, and why this method solves it. For example, the paragraph starting at line 50 argues for the SIA by saying that it will fully take advantage of the distributional return modeled by the DGN. This has two problems: 1. It is not true. The noise in the DGN is averaged out in the policy update such that the new policy is based only on the expected value of each action (Equation 13). 2. It does not connect back all the way to the end objective of the paper. If the goal of the method is to have faster policy learning due to better exploration, argue for that, and motivate why you think this is the right way to do it. Right now the motivation for the paper reads like you simply wanted to combine a few things; it would be much stronger to say precisely why.

[Author Response · NeurIPS 2020]

We thank the reviewers for their valuable comments and suggestions, which will help us to improve the paper. We first
respond to **R1**: **1)** While "twin-delay" is well-established to help estimate $Q(s, a)$, IDAC advances it to a distributional
RL setting for continuous actions to estimate the distribution of the discounted cumulative return $Z(s, a)$, whose
expectation is $Q(s, a)$. While the usual "twin-delay" only involves minimization of two scalars, the one in IDAC
is distinct in involving *element-wise* minimization of two *sorted* vectors, whose elements are $iid$ sampled from two
different D$G$Ns. Besides, we introduce SIA and an asymptotic lower bound for entropy estimation. Both new techniques
are helpful for IDAC to outperform the SOTA algorithms (please refer to Fig. 3). **2)** We will add the suggested ablation
studies. **3)** Similar to Fig. 2(a)(b) illustrating SIA at an early training stage, we have examined SIA at a late stage and
found that its differences from a diagonal Gaussian remain evident, as verified with both a normality test on the marginal
of each dimension of the SIA policy and the Pearson correlation test on many randomly selected action dimension pairs.
In addition, please see a related response to R3 (Lines 28-39) that strengthens the claim that SIA can represent more
complicated policy distributions. These details will be added. **4)** As suggested, we will draw more explicit distinctions,
such as the unique operation of sorting followed by element-wise minimization, and expand related work.

**R2**: **1)** We will clarify sorting is applied to each individual vector. We denote $\overrightarrow{x}_i$, which is a scalar, as the $i$-th element
after sorting vector $x$. **2)** We will better discuss related work to help avoid confusions and highlight our contributions.

**R3**: **1)** We are puzzled why IDAC is considered by R3 to be not sufficiently compared against other similar approaches;
we would appreciate R3 pointing out these missing baselines. First, from the actor-critic perspective, we have already
compared IDAC to SOTAs: PPO, SAC, & TD3. Second, from the distributional RL perspective, while there exist
well-known algorithms (e.g., C51, QR-D$Q$N, and IQN) for discrete actions, we find D4PG to be the only published
one designed for continuous controls. Note we did not add a direct comparison to D4PG (as well as SDPG) as its
implementation has notable differences from the Stable-baselines used in this paper. In particular, D4PG uses distributed
agents to do distributed sampling for the replay buffer, allowing an implied advantage in observing more state-action
pairs given the same number of policy gradient update steps. This is the main reason that we have adapted SDGP,
which improves over D4PG, into Stable-baselines and used it for ablation study. To help address R3's concern, we
will add these SDGP results (clearly worse than IDAC) into Fig. 1. Moreover, we have run the original D4PG code
in hope to further eliminate this concern, and found that it consistently underperforms IDAC given the same number
of policy gradient update steps. E.g., the max average returns of [D4PG, IDAC] after $10^6$ policy gradient steps are
$[9776 \pm 739, \mathbf{12222 \pm 157}]$ on HalfCheetah-v2, and $[4742 \pm 1320, \mathbf{5386 \pm 335}]$ on Walker2d-v2. **2)** *Verifying that*
*SIA would improve exploration*: We note good exploration by SIA can be implied from Fig. 3, by better empirical
performance; as the reward mechanism in continuous control RL tasks is complicated, such empirical comparison is
a common way to indirectly evaluate whether better explorations have been achieved for these tasks (e.g. Sec. 4.4
of Hong et al. [2018], arXiv:1802.04564). To more directly verify that SIA does improve exploration, we mimic
Sec. 5.1 of Haarnoja et al. [2017], arXiv:1702.08165 to introduce a Multigoal Environment with four equally-spaced
and well-separated destinations in a 2D map that provide large rewards of [200, 200, 200, 600], with spatial location-
dependent negative rewards elsewhere. This task requires extensive exploration to reach the optimal solution. We
combine a Gaussian actor or a SIA with an Actor-Critic (A2C) algorithm; each algorithm is trained with 10 independent
runs ($10^5$ episodes, one evaluation). In a single run, Gaussian often only explores no more than two (sub-optimal)
destinations, achieving average cumulative reward as $123.17 \pm 156.86$, while SIA in general successfully explores all
four destinations, achieving $\mathbf{336.84 \pm 26.9}$. These details verify that SIA does help exploration. **3)** We'd like to point
out that IDAC uses Huber loss, and it is unclear whether additional theoretical justifications are needed to combine
Huber loss with empirical Wasserstein distance. As quantile regression Huber loss is successfully used by QR-D$Q$N for
discrete actions, avoiding the potential issue of having biased gradients, we made the safe choice of using the same loss
in IDAC. We agree it would be an interesting future work to investigate the use of empirical Wasserstein distance (with
Huber loss) under IDAC. **4)** We will add more textual descriptions to improve our pseudo-code in the Appendix.

**R4**: **1)** We integrate D$G$N and SIA to allow them to help each other: *(i)* Modeling $G$ exploits distributional information
to help better estimate its mean $Q$ (note C51, which outperforms D$Q$N by exploiting distributional information, also
conducts its argmax operation on $Q$); *(ii)* A more flexible policy may become more necessary given a better estimated
$Q$. We have followed your suggestions to conducted additional ablation studies, which show removing either SIA or
D$G$N from IDAC in general negatively impacts its performance. These results will be added into revision. **2)** Quantile
regression loss can be applied to both explicit (as in QR-D$Q$N) and implicit distribution (as in D$G$N). We favor D$G$N
because it not only is more flexible, but also avoids a potential pitfall: QR-D$Q$N fixes the quantile locations and feeds
each $(s, a)$ to a deep neural network (NN) to estimate their corresponding values; A clear concern, however, is that
given input $(s, a)$, the NN output at a designated higher quantile is not guaranteed to be larger than that at a lower
quantile. By contrast, there is no such concern in D$G$N, which simply sorts its $iid$ sampled values. **3)** We clarify that
the SAC results were obtained by using the modern version [41] with the learned temperature. **4)** In Table 1 caption, we
will make the change to use "max average returns", obtained by taking the maximal of averaged evaluation score (over
4 random seeds). **5)** The normalizing integral does not contribute to the gradient of $\theta$ and hence omitted. In Eq(15), it is
$\pi_\theta(a^{(1)}|s, \xi^{(\ell)})$ rather than $\pi_\theta(a^{(\ell)}|s, \xi^{(\ell)})$, as indicated in Eq(12) multiple auxiliary variables $\xi$ are needed for each $a$.

[Meta-Review · NeurIPS 2020]

There was much discussion around the relationship with Tessler et al., which at first seemed quite close. I reached out to the authors for a clarification, as Reviewer 3 had omitted requesting it in their review. After this clarification, the reviewers decided that the work was indeed sufficiently novel and interesting. For the record, the reviewers unanimously appreciated receiving the author clarification (as, I can imagine, the authors appreciated sending it). However, that clarification itself was quite long (5 paragraphs). As I had not requested a specific word limit, I will take the blame for this, but I do encourage the authors to reply more tersely in the future, in the interest of fairness. Please consider the reviewers' updated reviews in your revisions.